# Ribosome structures to near-atomic resolution from thirty thousand cryo-EM particles

Xiao-chen Bai, Israel S Fernandez, Greg McMullan, Sjors HW Scheres*

Medical Research Council Laboratory of Molecular Biology, Cambridge, United Kingdom

**Abstract** Although electron cryo-microscopy (cryo-EM) single-particle analysis has become an important tool for structural biology of large and flexible macro-molecular assemblies, the technique has not yet reached its full potential. Besides fundamental limits imposed by radiation damage, poor detectors and beam-induced sample movement have been shown to degrade attainable resolutions. A new generation of direct electron detectors may ameliorate both effects. Apart from exhibiting improved signal-to-noise performance, these cameras are also fast enough to follow particle movements during electron irradiation. Here, we assess the potentials of this technology for cryo-EM structure determination. Using a newly developed statistical movie processing approach to compensate for beam-induced movement, we show that ribosome reconstructions with unprecedented resolutions may be calculated from almost two orders of magnitude fewer particles than used previously. Therefore, this methodology may expand the scope of high-resolution cryo-EM to a broad range of biological specimens.

## Introduction

For many years, electron cryo-microscopy (cryo-EM) has held the promise that macromolecular structure determination should in principle be possible to near-atomic resolution from only several thousand projection images of isolated particles in ice (*Henderson, 1995*; *Glaeser, 1999*). However, cryo-EM single-particle reconstructions in which protein main-chains may be traced reliably or individual side-chains are well resolved have been limited to large icosahedral viruses, and have typically required averaging over millions of asymmetric units (*Grigorieff and Harrison, 2011*). These results are in accordance with early observations that the contrast in EM images is not as good as expected from theory (*Henderson, 1995*). The main limiting factor of the information content in cryo-EM data is radiation damage: the electron dose needs to be limited to prevent the molecules from disintegrating while they are being imaged. However, radiation damage alone is not enough to account for the loss in expected contrast. Two additional factors that have been attributed to information loss in cryo-EM images are poor detective quantum efficiency (DQE, a frequency-dependent measure for the signal-to-noise performance) of conventional recording devices, and beam-induced movement or charging of the specimen during imaging (*Henderson, 1992*).

Traditionally, EM images have been recorded on photographic film, which has a large field of view and a reasonably good DQE, but is not convenient for high-throughput methods due to laborious steps of film development and digitization. Therefore, over the past decade many electron microscopes have been equipped with charge-coupled device (CCD) cameras. These digital detectors allow instant inspection of the images and are much more convenient in automated data collection schemes, so that large quantities of data may be obtained with relatively little effort (*Stagg et al., 2006*). However, the conversion of electrons into visible light that is detected by the CCD comes at the expense of a poorer DQE, in particular at higher voltages where many microscopes operate best. Consequently, despite the advantages of CCDs in terms of ease-of-use and data quantity, most of the

*For correspondence: scheres@mrc-lmb.cam.ac.uk

**Competing interests:** The authors declare that no competing interests exist

**Reviewing editor**: Werner Kühlbrandt, Max Planck Institute for Biophysics, Germany

**eLife digest** Determining the structure of proteins and other biomolecules at the atomic level is central to understanding many aspects of biology. X-ray crystallography is the best-known technique for structural biology but, as the name suggests, it works only with samples that can be crystallized. Electron cryo-microscopy (cryo-EM) could, potentially, be used to determine the atomic structures of biomolecules that cannot be crystallized, but at present the resolution that can be achieved with this approach is sufficient only for imaging certain types of viruses.

In cryo-EM, a solution of the biomolecule of interest is frozen in a thin layer of ice, and this layer is imaged in an electron microscope. By combining images of many identical biomolecules in many different orientations, it is possible to work backwards and determine their 3D structure. However, in order to determine this structure at high resolution, it is necessary to make repeated measurements to reduce high levels of noise in the images.

Cryo-EM images are usually recorded on a photographic film or a CCD (charge-coupled device) camera. However, photographic film is unsuitable for high-throughput methods because it has to be handled manually, while the efficiency of CCD cameras is limited because the electrons have to be converted into visible light to be detected. Digital cameras that can detect electrons directly have become available recently, and these are more efficient than both film and CCD cameras. They are also much faster, which means that it is possible to record videos of the sample during the time (typically ~1 s) it is being exposed to the electron beam. Processing these videos could then—in theory—compensate for any movements of the biomolecules that are induced by the electron beam. Along with radiation damage caused by the electrons, these beam-induced movements have been a major limitation on the resolution that can be achieved with cryo-EM.

Bai et al. demonstrate the potential of direct-electron detectors in cryo-EM by determining the structures of two ribosomes. Using a novel statistical algorithm to accurately follow the movements of the ribosomes during the time they are exposed to the electron beam, they are able to compensate for these movements, and this makes it possible to determine the structures of the ribosomes with near-atomic precision. Moreover, the resolution they achieve with just ~30,000 ribosomes is better than that previously achieved with more than a million ribosomes, allowing small details inside the ribosome – such as ß-strands and bulky amino-acid side chains – to be resolved with cryo-EM for the first time. The work of Bai et al. could, therefore, allow researchers to use cryo-EM to determine the structure of many more biomolecules with atomic precision.

near-atomic resolution cryo-EM structures published to date have been recorded on photographic film (*Grigorieff and Harrison, 2011*).

The second factor that has been attributed to the missing contrast in cryo-EM images is related to the radiation damage itself. For every elastic scattering event, which contributes positively to the phase contrast image, also three to four inelastic scattering events occur (*Henderson, 1995*). The inelastic events deposit energy in the sample, which leads to radiation damage by breaking covalent bonds and the generation of free radicals. Besides destroying the very structures one aims to determine, these interactions also generate charges in the sample that could lead to image blurring by deflecting the incoming electrons. In addition, radiolysis products such as hydrogen gas may build up internal pressure, thus leading to mechanical stress in the sample (*Leapman and Sun, 1995*). At high electron doses this stress ultimately leads to bubbling of the sample, while at much lower doses it has been postulated to induce movements that cause blurring of the images (*Glaeser, 2008*).

Recent technological advances may mitigate the image-blurring effects of both poor detectors and beam-induced movements. Building on technological developments that were initiated almost a decade ago (*Prydderch et al., 2003*), three different companies now sell digital cameras that detect electrons directly, that is, without the need to first convert electrons into visible light. Initial characterization of these direct electron detectors indicated that their DQE at high resolution is superior to that of CCD and film (*McMullan et al., 2009a*), in particular when backscattering of electrons is reduced by back-thinning of the substrate (*McMullan et al., 2009c*). An additional advantage of these devices is that they record images at high rates: ranging from 16 to 400 images per second for the currently available products. This allows the recording of a video during typical exposures, compared

to a single image on film or CCD. By processing these videos, Grigorieff and colleagues previously showed that beam-induced movements could be followed for large icosahedral viruses, although the gain in resolution compared to similar data recorded on film was not as large as expected, possibly due to experimental design (*Brilot et al., 2012*; *Campbell et al., 2012*).

Here, we assess the potential of a back-thinned FEI Falcon direct electron detector for cryo-EM structure determination by recording videos on two well-characterized test samples: the prokaryotic and eukaryotic ribosome. Ribosome samples have been characterized by cryo-EM for more than 20 years (*Frank et al., 1991*), and encompass characteristics that are favorable to cryo-EM structure determination: their large content of dense RNA and their large molecular weight (2.8 MDa for prokaryotic 70S and ~4 MDa for eukaryotic 80S ribosomes) result in images of relatively high contrast compared to many other macromolecular complexes, and RNA is known to be less radiation sensitive than protein. On the other hand, the ribosome is an intrinsically flexible macromolecular machine, which often leads to the presence of more than one structural state in ribosome samples. This represents a challenge, as advanced classification techniques are required to separate the different conformations into structurally homogeneous subsets that are amenable to single-particle recon-struction (*Scheres et al., 2007*; *Fischer et al., 2010*). In addition, the absence of symmetry in the ribosome makes it necessary to image many more particles (and to determine many more orientations) than is the case for icosahedral viruses with 60-fold symmetry. The combination of structural variability and the lack of symmetry may account for the fact that near-atomic resolution cryo-EM maps of ribosome structures have remained elusive thus far.

In the experiments described below, we used a 70S *T. thermophilus* ribosome sample to develop image-processing procedures that account for beam-induced movement of individual particles. Our detector operates at a rate of 17 frames per second, and we recorded 16-frame videos during 1-s exposures. At a total dose of 16 electrons/Å$^2$, each individual video frame therefore integrates less than 1 electron/Å$^2$. We expected the high noise levels in these individual frames to pose severe limitations on the accuracy with which these frames may be aligned. Because the alignment accuracy of individual particles remains unknown in typical single-particle analysis, we collected an entire data set as tilted pairs and used tilt-pair analysis to experimentally assess alignment errors for video frames (*Rosenthal and Henderson, 2003*). Guided by the results obtained from this analysis, we then developed a statistical video processing procedure that does not depend on tilted images. Application of this procedure to a second, untilted data set of 80S *S. cerevisiae* ribosomes serves to illustrate the impact these detectors, combined with suitable video processing procedures, will have on the field of cryo-EM structure determination.

## Results

We recorded tilt pairs of videos on the 70S *T. thermophilus* ribosome sample and used 2D and 3D classification to select a structurally homogeneous subpopulation of 15,202 particle pairs for 3D reconstruction (see 'Materials and methods'). In the first instance, we disregarded the time-dimension of the videos, and used 16-frame averages for each particle (*Figure 1*). By assessing the consistency between the orientations from independently refined tilt pair particles with the known microscope tilt geometry, we estimated an alignment precision for the tilt pairs of 2° (*Rosenthal and Henderson, 2003*) (*Figure 2A*). This is a significant improvement over the 4° precision reported for a similar sample recorded on CCD (*Henderson et al., 2011*). This improvement reflects the improved signal-to-noise ratios (SNRs) in the individual particles, which are a direct consequence of the improved DQE of the detector. The benefits of using particles with higher SNRs for structure determination are therefore twofold. On the one hand, averaging images with higher SNRs requires fewer particles to overcome the noise. On the other hand, higher SNRs result in smaller alignment errors, which then lead to decreased blurring in the reconstruction. Both effects will lead to higher-resolution maps from fewer particles, which explains the relatively high resolution of 6.5 Å that we obtained from only 15,202 particle pairs.

We then assessed how accurately particle movement may be followed during the electron exposure. To this purpose, we partitioned each original 16-frame average particle into multiple independent particles that were calculated as the average of a varying number of individual video frames. In this manner, we artificially created four additional data sets of increasing size, comprising averages of 2 × 8, 3 × 6, 4 × 4 or 8 × 2 video frames for each original particle. Tilt-pair analysis for these data showed that, as expected, the numbers of correctly aligned particles as well as the alignment

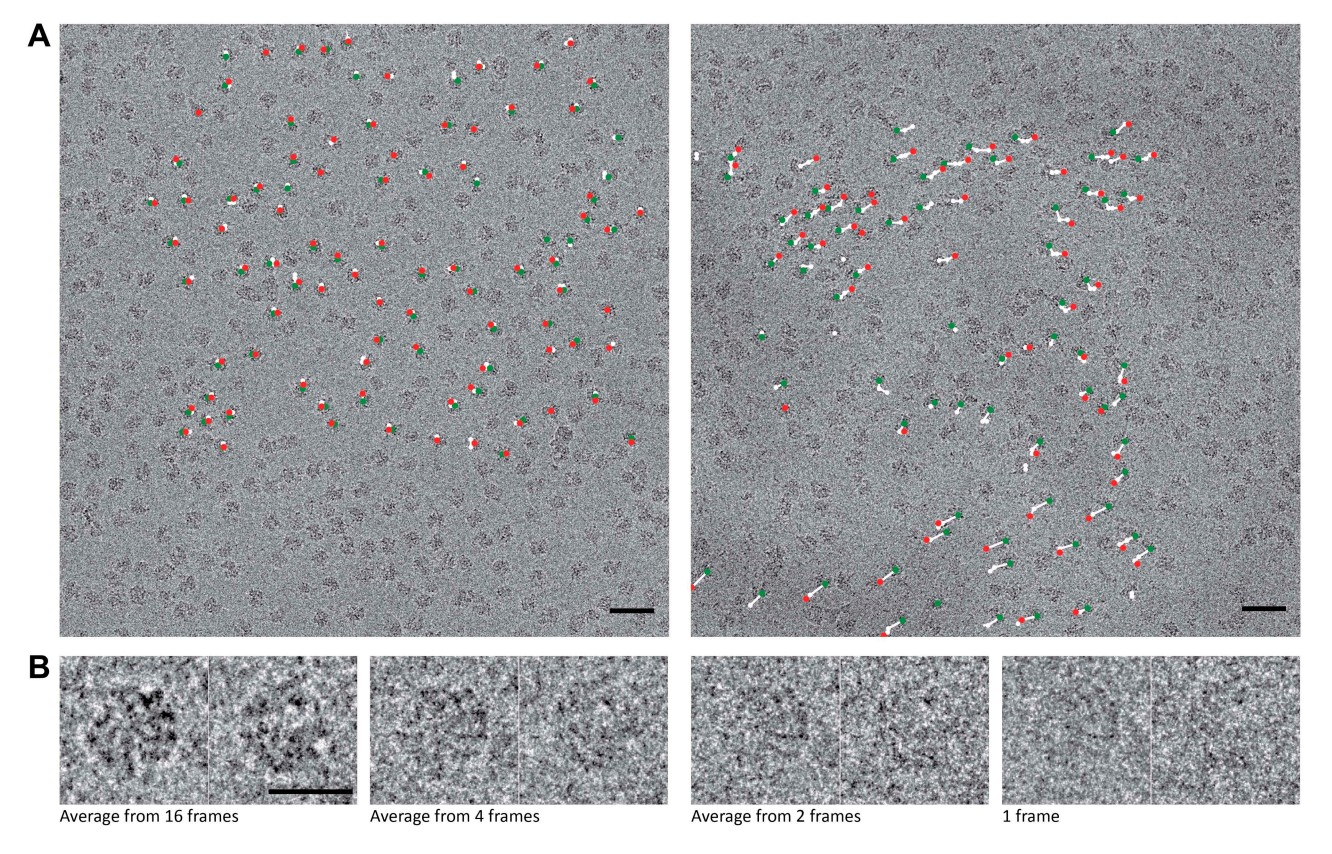

**A**

**B**

Average from 16 frames    Average from 4 frames    Average from 2 frames    1 frame

**Figure 1**. 70S data collected on a back-thinned FEI Falcon detector. (**A**) The 16-frame (1-s) average of two untilted videos. The scale bar indicates 50 nm. Relative positions for independently aligned four-frame averaged particles are shown with circles connected by white lines. The relative position of the average from the first four frames is shown in green, the relative position of the last four frames in red. The differences between these relative positions are exaggerated 25 times for improved clarity, and only those four-frame averages for which correct alignment was confirmed by tilt-pair analysis are included. Movements in the area on the left are smaller (up to 1.5 Å) then in the area on the right (up to 10 Å). (**B**) Examples of two individual ribosome particles as averages over a decreasing number of video frames. The scale bar indicates 20 nm. Zoomed-in areas of micrographs, additional individual particles and reference-free 2D class averages for both the 70S and 80S data are shown in *Figure 1—figure supplement 1*.

The following figure supplements are available for figure 1:

**Figure supplement 1**. Part of a recorded micrograph and reference-free class averages for the 70S and 80S data sets.

precision decrease with averaging over fewer video frames (*Figure 2B*). In addition, we noted a trend for particles that were calculated from later video frames to align less accurately, most likely as a result of accumulated radiation damage. To assess the effects of these alignment errors on reconstruction quality, we also inspected the Fourier Shell Correlation (FSC) curves that were reported for the four refinements (*Figure 2C*). The best reconstructions, at 5.4 Å resolution, were obtained when using averages of six or four individual video frames, which apparently represent the optimum between accurately describing particle movement and not introducing too large alignment errors for these data. Analysis of the orientations for the four-frame averaged particles that were aligned correctly according to the tilt-pair analysis revealed that overall rotations and translations during the exposure are in the order of 1.7 ± 1.2° and 4.2 ± 2.3 Å, respectively. Complicated patterns of movement for particles on some of the illuminated areas (e.g., see white lines in *Figure 1A*) indicate that the observed movements were not merely the result of a drifting stage, but more likely to be attributed to, as yet poorly understood, beam-induced effects.

Although useful for tilt-pair analysis, treating the averages of multiple video frames as completely independent also left valuable information unused. The observations that beam-induced movements are relatively small, and that the 16-frame averages may be aligned most accurately, represent valuable

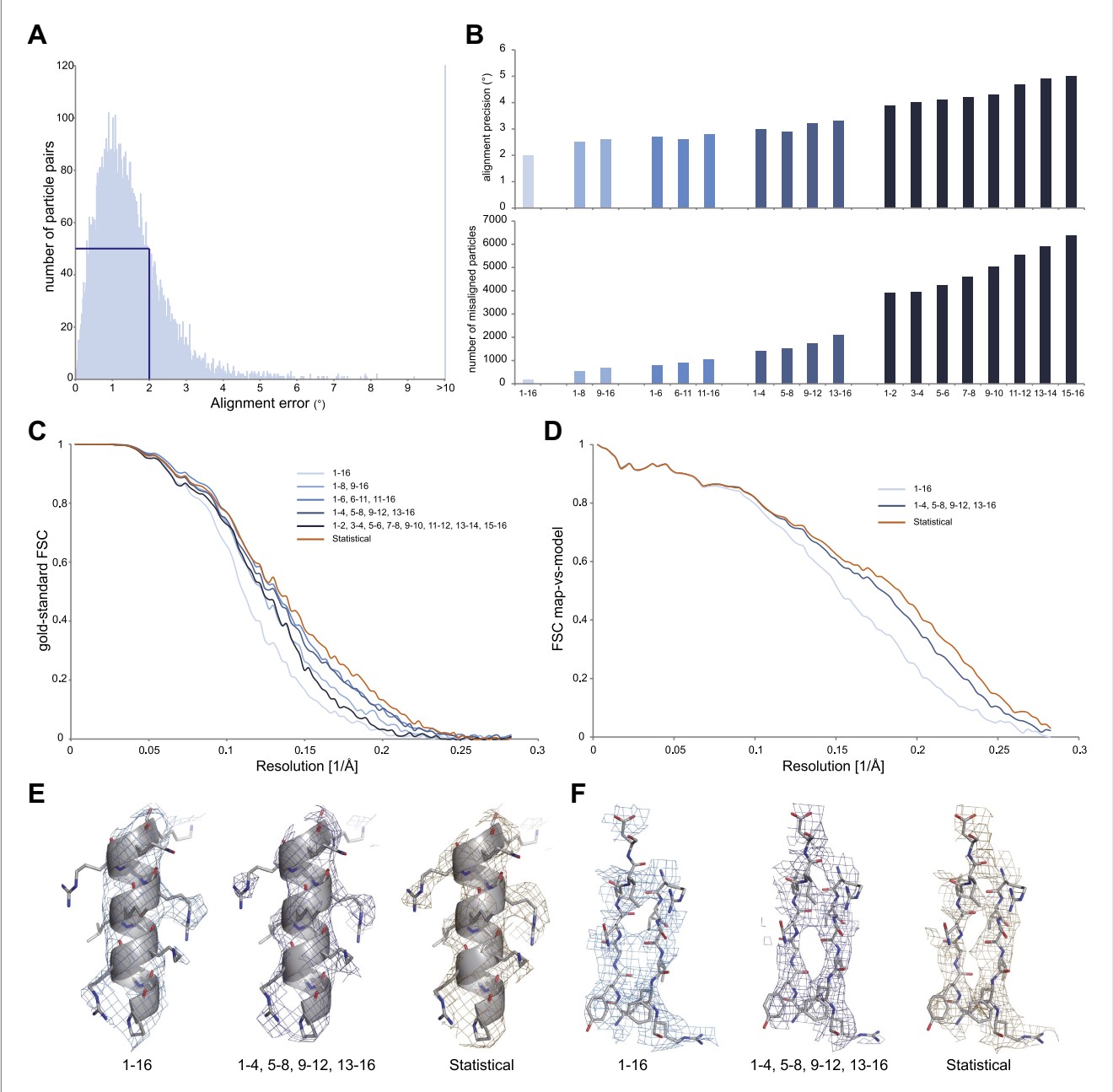

**Figure 2**. Development of video processing procedures on the 70S data set. (**A**) Histogram of tilt-pair alignment errors for particles that were calculated as 16-frame averages. The width of the first peak at half its height is 2°. This value is plotted as the tilt-pair alignment precision in (B). (**B**) Tilt-pair alignment precision (top) and the number of incorrectly aligned particle pairs (bottom) for independent refinements of 16-frame, 8-frame, 6-frame, 4-frame and 2-frame averages (ranges in frame numbers are indicated on the x-axis); the total number of particle pairs was 15,202. Particle pairs with alignment errors larger than three times the reported precision were considered as aligned incorrectly. (**C**) Gold-standard FSC curves. The same blue colors are used as in (B); orange lines indicate the results of the statistical video-processing approach described in the main text. (**D**) FSC-curves between a rigid-body fitted atomic model and the cryo-EM maps (using the same color scheme as in [C]). (**E–F**) Illustrative density and atomic model for the reconstructions obtained from 16-frame averaged particles (light blue), independently refined four-frame averages (dark blue) and the statistical video processing procedure (orange). The density maps were sharpened with B-factors of −211, −185 and −178 Å², respectively. Complete density maps for these three reconstructions are shown in **Figure 2—figure supplement 1**.

The following figure supplements are available for figure 2:

**Figure supplement 1**. Overall views of the 70S ribosome reconstructions obtained from 16-frame averaged particles (left), independently refined four-frame averages (middle) and the statistical video processing procedure (right).

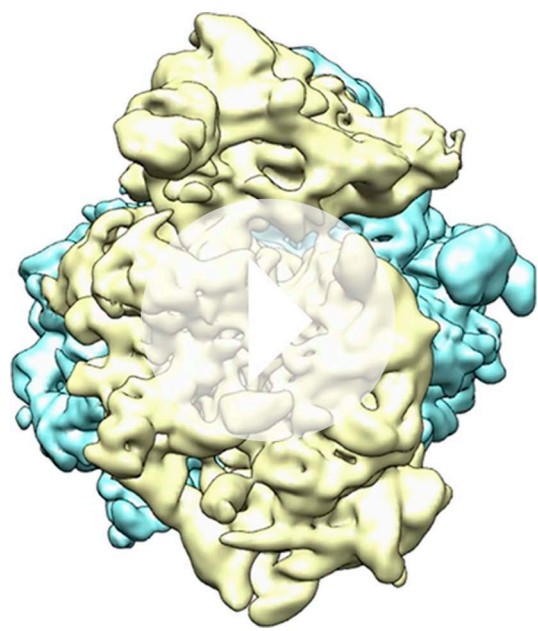

**Video 1**. Shown are the density maps for the two classes of the 80S data set after 3D classification, which display a difference of ~2° in ratchet-like rotation. The resolution is 6.5 Å for class 1 and 6.7 Å for class 2. However, the maps appear to be of lower resolution as they have not been sharpened.

prior knowledge that may be efficiently expressed using statistical refinement methods. In the 'Materials and methods' section we describe the implementation of a probabilistic prior on the orientations of the video frames inside a Bayesian refinement approach (*Scheres, 2012a*). This prior down-weights orientations for the video frames in a continuous manner according to their distance from the orientation determined for the 16-frame averages. Application of this procedure to the 70S ribosome data yielded a reconstruction with a resolution of 5.1 Å (orange line in *Figure 2C*). Comparison of the FSCs between the reconstructed maps and a rigid-body fitted crystal structure (*Selmer et al., 2006*), decreasing estimated B-factors, and visual inspection of the maps all confirmed that reconstruction quality may be improved by following the movement of individual particles, and that the statistical approach outperforms the straightforward use of multi-frame averaged particles in this respect (*Figure 2D–F*, *Figure 2–figure supplement 1*).

Finally, the developed procedure was applied to a second, untilted data set that was collected on an 80S *S.cerevisiae* ribosome sample during a single, manual microscopy session. Classification identified two major conformations, comprising 35,813 and 22,638 particles, respectively. A difference of ~2° in ratchet-like movement represents the largest difference between the two classes (also see *Video 1*). Using the statistical video processing procedure, refinement of class 1 yielded a map with an overall resolution of 4.5 Å, while class 2 yielded a 4.6 Å map (*Figure 3A*). However, both maps still exhibited signs of unresolved structural heterogeneity, in particular in the small subunit and most notably for class 1 (*Figure 3B*). Therefore, the overall resolution estimates are too optimistic for the disordered parts, whereas the most stable parts of the structures still contain significant information beyond the overall estimated resolutions. The density for the 60S subunit clearly shows separation of β-strands, the pitch of α-helices, density for many side chains, and the separation of RNA bases, indicating that useful information to ~4 Å resolution is present in the best parts of the map (*Figure 3C–G*, *Video 2*). The level of detail in the density for the 40S subunit varies, but is generally lower than for the 60S subunit (*Figure 3H,I*, *Video 2*). Calculation of FSC curves between the cryo-EM maps and rigid-body fitted crystal structures for the 60S and 40S subunits (*Ben-Shem et al., 2011*) confirmed the estimated resolutions, as well as the variation in quality of the density across the maps (*Figure 3J,K*).

## Discussion

The reconstructions presented in this paper are of significantly higher resolution than any prokaryotic or eukaryotic ribosome structures that were calculated from cryo-EM data on photographic film or CCD. Moreover, they were obtained from unprecedented small numbers of particles. For comparison, currently the highest resolution ribosome map in the EMDB (an 80S structure from the plant *Triticum aestivum*) was calculated from almost 1.4 million particles and was reported to be at 5.5 Å resolution (*Armache et al., 2010*). We obtained reconstructions with useful information up to ~4 Å resolution from nearly two orders of magnitude fewer particles. The observation that we could calculate maps with useful information up to 90% of the Nyquist frequency surpassed our initial expectations and clearly illustrates the potential of direct electron detectors combined with adequate video processing for cryo-EM structure determination.

Incorporation of the new detectors into automated data collection schemes will result in much larger amounts of data than we have used in our study. For relatively high molecular-weight complexes that display sufficient contrast for accurate alignment and classification, this may then allow

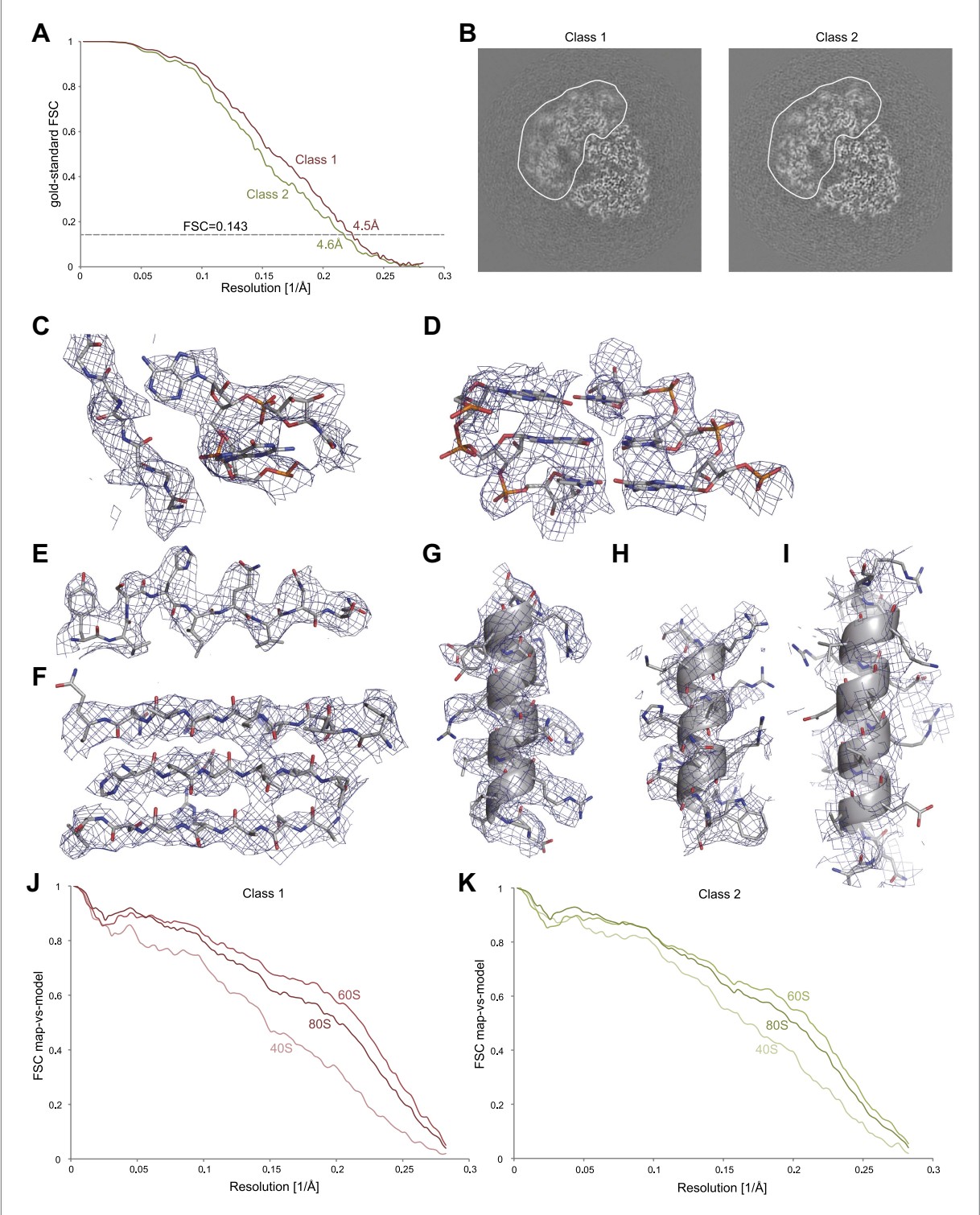

**Figure 3**. Application of the statistical video processing procedure to an 80S ribosome data set. (**A**) Gold-standard FSC curves for class 1 (red) and class 2 (green). (**B**) Slices through the reconstructions of class 1 (left) and class 2 (right). The fuzzy appearance for the density of the 40S subunits (green and red lines) is an indication of unresolved structural heterogeneity. (**C–G**) Densities for the 60S subunit of class 1 showing a protein loop interacting with a flipped-out RNA base (**C**), a short stretch of an RNA helix (**D**), a β-strand (**E**), a β-sheet (**F**), and an α-helix (**G**). (**H–I**) Density for the 40S subunit of class 1 showing a well-resolved α-helix (**H**) and a poorly-resolved one (**I**). The density map of class 1 was sharpened with a B-factor of −160 Å². (**J**) FSC curves between the map of class 1 and the rigid-body fitted atomic models of the entire 80S particle, and for the 40S and 60S subunits separately. (**K**) As in J, but for class 2.

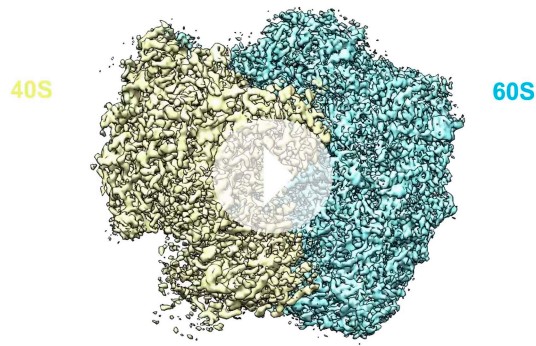

**Video 2**. Shown is the cryo-EM density map for class 1 of the 80S data set together with the atomic models that are also shown in **Figure 3C–I**. Density for the 60S subunit is shown in blue, density for the 40S subunit in yellow. The density for the 40S subunit in the overall view is filtered to 5.0 Å resolution for improved clarity, all other densities are filtered at 4.1 Å resolution.

high-resolution reconstruction of many more structural states, or of states that are adopted by only a small fraction of the particles. Moreover, if the sample is sufficiently rigid to exist in only one or a few distinct states, then larger amounts of data are expected to lead to even higher resolutions than the ones reported here. For example, following the SNR considerations proposed by (*Rosenthal and Henderson, 2003*), our overall resolution of 4.5 Å would increase to ~3.3 Å if we would apply our approach to 1 million particles of similar quality (provided no other resolution-degrading factors play a role). Therefore, we foresee that the combination of direct electron detectors, video processing approaches, and automated data collection schemes will significantly increase the number of specimens for which de novo building of atomic models into cryo-EM maps becomes feasible.

The advantages of direct electron detectors and video processing procedures are not limited to relatively large complexes alone. Current alignment procedures are often severely limited by low SNRs in images of complexes that are smaller than ~300 to 500 kDa. Also in this respect, the ability to record videos may provide significant benefits. Because SNRs in individual particles drop quickly with increasing spatial frequency image alignments are typically mainly driven by the lower-resolution components in the images (*Henderson et al., 2011*; *Scheres and Chen, 2012*). These components have been observed to remain intact after much higher electron doses than the high-resolution information (*Hayward and Glaeser, 1979*; *Baker and Rubinstein, 2010*). Therefore, relatively long videos with a high dose could be used to obtain reliable low-resolution components, while a dose-dependent weighting scheme to account for radiation damage may be used to optimize the SNR for higher spatial frequencies at the same time (*Campbell et al., 2012*). Such procedures are expected to significantly lower the size limit of particles that may still be aligned correctly and are thus amenable to cryo-EM structure determination. Still, accurately following the movement of small particles during videos will be more challenging than for the high-contrast ribosome particles presented here. Therefore, in order to also obtain near-atomic resolutions for smaller particles, further investigations into the nature of beam-induced image blurring and how to stop it will still be necessary. Also these investigations are expected to benefit significantly from the possibility to analyze beam-induced movements during videos recorded by direct electron detectors.

Based on our observations and the potential for further improvements in the technique, we believe the future for cryo-EM structure determination to be bright. Currently available detectors, combined with video processing algorithms like the one presented here, will result in higher-resolution cryo-EM structures for many more samples than those reported until now. Meanwhile, cryo-EM image quality will keep improving as ongoing technological developments such as single-electron counting detectors (*McMullan et al., 2009b*), phase-plate imaging (*Nagayama, 2011*) and better sample preparation techniques mature. Therefore, we are optimistic that the field will continue to progress towards fulfilling the promise of providing near-atomic resolution reconstructions, and thereby more detailed biological insights, for a wide range of specimens.

## Materials and methods

### Sample preparation

70S *T. thermophilus* ribosome samples with tRNAs and mRNA were produced as described previously (*Selmer et al., 2006*). 80S ribosomes were purified from *S. cereviseae* strain YAS-2488. Cells were grown to $OD_{600}$ = 2–4 and starved for 10 min at 4°C in buffer without sucrose (20 mM Hepes-KOH pH 7.45, 150 mM KCl, 150 mM K-acetate, 10 mM Mg-acetate, 1 mg/ml heparin, 0.1 mM PMFS, 0.1 mM benzamidine, 2 mM DTT). After starvation, cells were frozen in liquid nitrogen and mechanically disrupted by a blender machine operated under liquid nitrogen conditions. The lysate was allowed to

defrost at 4°C and then clarified by a 20 min centrifugation at 14,500×g. The 80S ribosomes from the supernatant were pelleted through a sucrose cushion for 4 hr at 45,000 rpm in a Ti45 Beckman Coulter rotor (Palo Alto, California, USA), in buffer that was supplemented with 1 M sucrose. The pellets were re-suspended in a sucrose gradient buffer without sucrose (20 mM Hepes-KOH pH 7.45, 50 mM KCl, 5 mM Mg-acetate, 0.1 mM PMFS, 0.1 mM benzamidine, 2 mM DTT) and incubated for 15 min with 1 mM puromycin. The sample was loaded on a 10–40% sucrose gradient and centrifuged for 18 hr at 28,000 rpm in a Ti25 zonal rotor. A single peak after gradient fractionation was confirmed to correspond to 80S ribosome particles. All ribosomal proteins, as well as the protein Stm1, were identified using mass spectrometry. For storage in liquid nitrogen, the buffer was exchanged to buffer M (3 mM Hepes-KOH pH 7.45, 6.6 mM Tris-acetate pH 7.2, 3 mM $NH_4Cl$, 6.6 mM $NH_4$-acetate, 48 mM K-acetate, 4 mM Mg-acetate, 2.4 mM DTT) and the sample was concentrated to 6 µM.

## Electron microscopy

For both the 70S and 80S samples, aliquots of 3 µl at a concentration of ~80 nM were incubated for 30 s on glow-discharged holey carbon grids (Quantifoil R2/2), on which a home-made continuous carbon film (estimated to be ~30-Å thick) had previously been deposited. Grids were blotted for 2.5 s and plunge-frozen in liquid ethane using an FEI Vitrobot. Grids were transferred to an FEI Polara G2 microscope that was operated at 300 kV. A C2 aperture of 70 µm and an objective aperture of 100 µm were used. Defocus was varied from 1.3 to 3.8 µm. Using an extraction voltage of 3900 V, a gun lens setting of 2 and a spotsize of 4 or 5, an estimated dose of 16 electrons/$Å^2$ was applied during 1-s exposures. The beam used was larger than the Quantifoil hole, illuminating the carbon all around the hole. Images were recorded at the approximate center of the hole on a back-thinned FEI Falcon detector at a calibrated magnification of 79,096 (yielding a pixel size of 1.77 Å). The small area that was imaged relative to the area that was illuminated resulted in a beam-tilt that was much smaller than the one expected from the relatively large C2 aperture that we used, but we cannot exclude that our final resolution was limited by beam-tilt. An in-house built system was used to intercept the videos from the detector (we were capable of recording 16 frames during a 1-s exposure). All data were collected manually during two half-day sessions for the 70S sample, and a single half-day session for the 80S sample. Tilt pairs were collected at tilt angles of 0° and 10°. Electron micrographs were evaluated for astigmatism and drift. For the 70S sample, 159 out of 285 micrograph pairs were selected for further analysis; 260 out of 291 micrographs were selected for the 80S sample.

## Image processing

A total of 24,044 70S particle pairs and 72,447 80S particles were selected using the *swarm* tool in the *e2boxer.py* program of EMAN2 (*Tang et al., 2007*), contrast transfer function parameters were estimated using CTFFIND3 (*Mindell and Grigorieff, 2003*), and all 2D and 3D classifications and refinements were performed in RELION (*Scheres, 2012b*). Prior to 3D refinement, both data sets were subjected to reference-free 2D class averaging and 3D classification to identify structurally homogeneous subsets. Initial 3D classifications were run for 25 iterations, with four classes, at the original image size, with an angular sampling of 7.5°, and a regularization parameter $T = 4$. For the 70S data set, discarded particles included 50S subunits and 70S ribosomes with tRNAs in the E- and P-sites, and 15,202 particles were selected to correspond to 70S ribosomes with a single tRNA in the P-site. For the 80S data, dissociated subunits could again be identified in the 2D class averages, but initial 3D classification did not yield structurally distinct classes. However, after 3D refinement of a single model, blurry density for the 40S subunit indicated that significant structural heterogeneity was still present. In a subsequent 3D classification run with four classes, an angular sampling of 1.8° was combined with local angular searches around the refined orientations, and the refined single model was used as a starting model. This calculation separated two conformations with different degrees of ratchet-like movement, comprising 35,813 and 22,638 particles, respectively.

Initial 3D classifications and all 3D refinements were started from ribosome maps that were downloaded from the Electron Microscopy Data Bank (EMDB-1657 for the 70S; EMDB-1780 for the 80S ribosome; *Seidelt et al., 2009*; *Armache et al., 2010*) and subsequently low-pass filtered to 60 Å. All 3D refinements used gold-standard FSC calculations to avoid overfitting and reported resolutions were based on the FSC = 0.143 criterion (*Scheres and Chen, 2012*). Final FSC curves were calculated using a soft spherical mask (with a 5-pixel fall-off) on the two independent reconstructions. Prior to visualization, all density maps were corrected for the modulation transfer function (MTF) of the

detector, and then sharpened by applying a negative B-factor that was estimated using automated procedures (*Rosenthal and Henderson, 2003*).

Rigid body fitting of 70S and 80S crystal structures (PDB-IDs: 2WH1-2, *Weixlbaumer et al., 2008*, and 3U5B-E, *Ben-Shem et al., 2011*, respectively) was performed using UCSF Chimera (*Pettersen et al., 2004*). In order to describe the distinct degrees of ratchet rotation in the two 80S conformations, the 40S and 60S subunits were fitted separately. The resulting models merely served to illustrate the quality of our maps; we did not analyze our maps in terms of differences with the crystal structures. To allow others to mine our structures for additional information (e.g., we see continuous density for RNA nucleotides 440–499 of expansion segment 7 and nucleotides 1954–2093 of expansion segment 27 in 25S, which were not modelled in the 80S crystal structure), we have deposited our maps at the EMDB (accession numbers 2275, 2276 and 2277).

## Statistical video processing

A novel procedure for video-frame alignment was developed that exploits the relatively high accuracy of aligning 16-frame average particles (*Figure 2B*), as well as the prior knowledge that particles are unlikely to undergo very large rotations or translations during the 1-s exposure. To this purpose, we defined Gaussian prior distributions on the rotations and translations of the video frames, and centered these distributions at the observed orientations for alignments with the corresponding 16-frame average particles. The priors were then incorporated as prior probability distributions, $P(\varphi/\Theta,Y)$ in Eq. 7 of *Scheres (2012a)*, in the Bayesian refinement approach of the RELION program. This open-source program may be downloaded from http://www2.mrc-lmb.cam.ac.uk/relion; the video-processing procedures described here are available in version 1.2. The widths of the priors were implemented as user-controlled parameters that may be tuned to express the expected amount of movement during the video. In addition to the Gaussian priors, we also implemented an option to use running averages of a user-defined number of frames for the alignments. This allows more precise sampling of the particle movement than the four discrete 4-frame averages along the 16-frame videos shown in *Figure 1*. The corresponding orientations (or probability distributions in the Bayesian approach) are only applied to the single, middle video frame of the running average window. For all calculations shown in this paper, we used running averages of five video frames; a standard deviation of 1° for the priors on the Euler angles; and the standard deviation that was estimated for the picked particle positions in the 16-frame alignment for the prior on the translations. Rotational searches (or integrations in the Bayesian approach) were then limited to ±3° with a step size of 0.45°, while translational searches were performed up to ±2 pixels with a step size of 0.5 pixels. We note that in this case we used the observed translations and rotations of the four-frame average refinements to choose the widths of our priors. Alternatively, one may optimize these values based on their effects on the gold-standard FSC curve. Although not pursued in this study, the Bayesian approach also provides a convenient framework to estimate these widths from the data themselves. In addition, as the field gains a better understanding of beam-induced movements, more detailed priors (e.g., with widths that depend on the accumulated dose for each frame) may easily be incorporated.

## Acknowledgements

We are grateful to Shaoxia Chen, Toby Darling, Jake Grimmett and Ann Kelley for technical support, and to Richard Henderson for helpful discussions. Ribosome samples were prepared in the laboratory of Venki Ramakrishnan, who is funded by the UK Medical Research Council (MRC) through grant U105184332 and the Wellcome Trust.

## Additional information

### Funding

| Funder | Grant reference number | Author |
| --- | --- | --- |
| UK Medical Research Council | MC_UP_A025_1013 | Sjors HW Scheres |
| Wellcome Trust | | Israel S Fernandez |
| UK Medical Research Council | U105184332 | Israel S Fernandez |

The funders had no role in study design, data collection and interpretation, or the decision to submit the work for publication.

## Author contributions

XB, Acquisition of data, Analysis and interpretation of data; ISF, Analysis and interpretation of data, Contributed unpublished essential data or reagents; GM, Acquisition of data; SHWS, Conception and design, Analysis and interpretation of data, Drafting or revising the article

## Additional files

### Major datasets

The following datasets were used:

| Author(s) | Year | Dataset title | Dataset ID and/or URL | Database, license, and accessibility information |
|---|---|---|---|---|
| Bai XC, Fernandez IS, McMullan G, Scheres SHW | 2013 | Ribosome structures to near-atomic resolution from thirty thousand cryo-EM particles: *S. cereviseae* 80S ribosome | 2275; http://www.ebi.ac.uk/pdbe-srv/emsearch/atlas/2275_summary.html | In the public domain at the Electron Microscopy Data Bank (EMDB) at PDBE: http://www.ebi.ac.uk/pdbe/emdb/ |
| Bai XC, Fernandez IS, McMullan G, Scheres SHW | 2013 | Ribosome structures to near-atomic resolution from thirty thousand cryo-EM particles: *S. cereviseae* 80S ribosome | 2276; http://www.ebi.ac.uk/pdbe-srv/emsearch/atlas/2276_sample.html | In the public domain at the Electron Microscopy Data Bank (EMDB) at PDBE: http://www.ebi.ac.uk/pdbe/emdb/ |
| Bai XC, Fernandez IS, McMullan G, Scheres SHW | 2013 | Ribosome structures to near-atomic resolution from thirty thousand cryo-EM particles: *T. thermophilus* 70S ribosome | 2277; http://www.ebi.ac.uk/pdbe-srv/emsearch/atlas/2277_summary.html | In the public domain at the Electron Microscopy Data Bank (EMDB) at PDBE: http://www.ebi.ac.uk/pdbe/emdb/ |
| Bai XC, Fernandez IS, McMullan G, Scheres SHW | 2013 | Ribosome structures to near-atomic resolution from thirty thousand cryo-EM particles: 80S micrograph movies | | Unprocessed micrograph movies for the 80S sample will be distributed as 3DEM test data by the EMDB. Meanwhile, these data are available from the authors upon request |

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
