## [Decision Letter]

Thank you for choosing to send your work entitled “Ribosome structures to near-atomic resolution from thirty thousand cryo-EM particles” for consideration at *eLife*. Your article has been evaluated by a Senior editor and 3 reviewers, one of whom is a member of our Board of Reviewing Editors.

The following individuals responsible for the peer review of your submission want to reveal their identity: Werner Kühlbrandt (Reviewing editor); Niko Grigorieff (peer reviewer). The Reviewing editor and the other reviewers discussed their comments, and the Reviewing editor has assembled the following summary to help you prepare a revised submission.

Bai et al report the cryo-EM structure of the yeast 80S ribosome at close to 4 Å resolution, determined from images recorded with the new direct electron detection Falcon camera. An essential modification of the camera readout that is not yet commercially available enabled the recording of dose-fractionation movies. By aligning and averaging individual subframes from these movies, the authors were able to overcome the effects of beam-induced specimen movement, which has been the most serious limitation in high-resolution data acquisition by cryo-EM until now. Remarkably, movies of 35,000 particles or less recorded in a single session were sufficient to achieve this impressive result.

Similar cameras have been developed by Gatan and the Direct Electron company. The high detective quantum efficiency of these innovative devices makes them far superior to the widely used scintillator-based CCD cameras and even to film, which has been the medium of choice for high-resolution cryo-EM data recording. The new direct electron detection cameras are set to revolutionize cryo-EM, and the present study is the first to take advantage of their full power to determine the structure of an important macromolecular complex at high resolution.

An important factor contributing to this success is new software. Scheres' powerful Bayesian approach already proved in the RELION program to excel in the job of sorting projections from heterogeneous samples. This frame-by-frame particle alignment had already been studied by Brilot et al (2012) and Campbell et al. (2012) but with limited improvement in resolution. Scheres and coworkers prove the superiority of their statistical framework over more traditional methods, bringing the resolution of single-particle methods into a new territory. The paper requires only minor modifications, as below.

1. The concept of a “probabilistic prior” will be unfamiliar to most readers and requires a brief explanation. How were the widths of the priors for the statistical movie processing chosen? Did the authors perform some kind of optimization, maybe by observing improvements in the gold-standard FSC? How did the authors take into account the observation that the beam-induced movement per dose unit is larger at the beginning of an exposure and smaller towards the end?

2. Comments on image processing:

* What was the beam tilt in the centre of the illuminated area, and how was it measured or estimated? How much beam tilt would be acceptable at 4 Å resolution? Can you exclude that the resolution was limited by beam tilt?

* Please provide details on the classification (window size, number of classes, regularization factor, how many cycles?).

* Code for this novel alignment procedure should be provided.

* It would be desirable, and a great asset for the community, if the data would be deposited as a benchmark that allowed the testing of other algorithms. EMDB now allows for such depositions.

3. Comments on figures:

* Figure 1: The red and green dots reinforce the (false) notion that the particle moves by this amount. The underlying vector with exaggerated length is okay.

* Figure 2G: there should be a panel 2G (or a separate figure) showing complete density maps of the 80S ribosome for the three modes of frame integration. Many readers are familiar with the appearance of the ribosome in various reconstructions and reproduction of the full maps would provide a good qualitative appreciation of the results.

---

## [Author Response]

*1. The concept of a “probabilistic prior” will be unfamiliar to most readers and requires a brief explanation. How were the widths of the priors for the statistical movie processing chosen? Did the authors perform some kind of optimization, maybe by observing improvements in the gold-standard FSC? How did the authors take into account the observation that the beam-induced movement per dose unit is larger at the beginning of an exposure and smaller towards the end*?

We added the following sentence to clarify the role of the prior: “This prior down-weights orientations for the movie frames in a continuous manner according to their distance from the orientation determined for the 16-frame averages.”

We did not do any optimization on the width of the prior, but rather used the information gained from the 4-frame average refinements. However, one could indeed use the gold-standard FSC curve to find a suitable width. More elegantly, in future work these widths could be estimated from the data themselves. We also did not account for different movements per dose-unit, although this could also be done. To reflect all this, we added the following text to the Materials and methods: “We note that in this case we used the observed translations and rotations of the 4-frame average refinements to choose the widths of our priors. Alternatively, one may optimize these values based on their effects on the gold-standard FSC curve. Although not pursued in this study, the Bayesian approach also provides a convenient framework to estimate these widths from the data themselves. In addition, as the field gains a better understanding of beam-induced movements, more detailed priors (e.g. with widths that depend on the accumulated dose for each frame) may easily be incorporated.”

*2. Comments on image processing*:

** What was the beam tilt in the centre of the illuminated area, and how was it measured or estimated? How much beam tilt would be acceptable at 4 Å resolution? Can you exclude that the resolution was limited by beam tilt*?

We did not measure beam-tilt, and cannot exclude that the resolution is limited by beam-tilt. To make this explicit, we modified the following sentence: “The small area that was imaged relative to the area that was illuminated resulted in a beam-tilt that was much smaller than the one expected from the relatively large C2 aperture that we used, but we cannot exclude that our final resolution was limited by beam-tilt.”

** Code for this novel alignment procedure should be provided*.

All our code is open-source. The 1.2 release of RELION is being prepared at the moment, but is already available for beta-testing upon request. The following sentence was added to the Materials and methods: “This open-source program may be downloaded from http://www2.mrc-lmb.cam.ac.uk/relion; the movie-processing procedures described here are available in version 1.2.”

** It would be desirable, and a great asset for the community, if the data would be deposited as a benchmark that allowed the testing of other algorithms. EMDB now allows for such depositions*.

This is a good idea. We have contacted EMDB about the possibility of hosting our 80S data set. Comprising ∼260Gb, data volume could be an issue.

*3. Comments on figures*:

** Figure 1: The red and green dots reinforce the (false) notion that the particle moves by this amount. The underlying vector with exaggerated length is okay*.

We modified the legend as follows: “The relative position of the average from the first four frames is shown in green, the relative position of the last four frames in red. The differences between these relative positions are exaggerated 25 times for improved clarity.”

** Figure 2G: there should be a panel 2G (or a separate figure) showing complete density maps of the 80S ribosome for the three modes of frame integration. Many readers are familiar with the appearance of the ribosome in various reconstructions and reproduction of the full maps would provide a good qualitative appreciation of the results*.

In our opinion, it is hard to appreciate the improvement in density at 5–4 Å resolution in the large views of complete ribosomes. Still, we understand that these views may be useful to cryo-EM ribosome experts as an overall quality check. Therefore, we have included the requested views as a Figure supplement to Figure 2.